# GROUP-WISE OPTIMIZATION FOR SELF-EXTENSIBLE CODEBOOKS IN VECTOR QUANTIZED MODELS

## ABSTRACT

Vector Quantized Variational Autoencoders (VQ-VAEs) leverage self-supervised learning through reconstruction tasks to represent continuous vectors using the closest vectors in a codebook. However, issues such as codebook collapse persist in the VQ model. To address these issues, existing approaches employ implicit static codebooks or jointly optimize the entire codebook, but these methods constrain the codebook's learning capability, leading to reduced reconstruction quality. In this paper, we propose Group-VQ, which performs group-wise optimization on the codebook. Each group is optimized independently, with joint optimization performed within groups. This approach improves the trade-off between codebook utilization and reconstruction performance. Additionally, we introduce a training-free codebook resampling method, allowing post-training adjustment of the codebook size. In image reconstruction experiments under various settings, Group-VQ demonstrates improved performance on reconstruction metrics. And the post-training codebook sampling method achieves the desired flexibility in adjusting the codebook size. The core code is available at `https://anonymous.4open.science/r/Group-VQ_anonymous-60E3`

## 1 INTRODUCTION

Vector Quantization (VQ) (Gray, 1984) is a technique that maps continuous features to discrete tokens. Specifically, VQ defines a finite codebook and selects the closest code vector for each feature vector by calculating a similarity measure, typically Euclidean distance or cosine similarity (Yu et al., 2021). This selected code vector serves as the discrete representation of the feature vector. VQ-VAE (Van Den Oord et al., 2017; Razavi et al., 2019) employs vector quantization as an image tokenizer, which quantizes the feature map output by the encoder to represent an image as a series of integer indices. The decoder then uses only the quantized feature map to reconstruct the image. Due to the non-differentiability (Huh et al., 2023) of the quantization operation, the Straight-Through Estimator (STE) (Bengio et al., 2013) enables the encoder to receive gradients from the task loss by copying the gradients of the quantized vectors to the pre-quantized vectors. VQ has found widespread applications in autoencoders (Van Den Oord et al., 2017; Razavi et al., 2019; Zhao et al., 2024a) and generative models (Rombach et al., 2022; Dhariwal et al., 2020; Tian et al., 2024; Weber et al., 2024; Yu et al., 2024a).

Despite achieving success in numerous applications, traditional VQ training often encounters the issue of low codebook utilization, where only a subset of code vectors are used and updated, leading to "codebook collapse" (Roy et al., 2018; Huh et al., 2023; Yu et al., 2024b), which limits the model's encoding capability. To address these challenges, various improvements have been proposed, such as reducing the dimensionality of the latent space (Yu et al., 2021; Mentzer et al., 2023; Yu et al., 2023), initializing the codebook with pretrained models (Huh et al., 2023; Zhu et al., 2024a), and jointly optimizing the entire codebook (Zhu et al., 2024b; Huh et al., 2023). In our paper, we refer to these methods as "Joint VQ". These methods have shown promising results, achieving near 100% codebook utilization. However, in order to reach a 100% utilization rate, our experiments indicate that these methods have restricted the learning ability of the codebook to some extent, resulting in performance differences under the same utilization rate.

To alleviate this issue, we propose to approach codebook optimization from a group perspective, thereby naturally introducing the Group-VQ method. This method organizes the codebook into

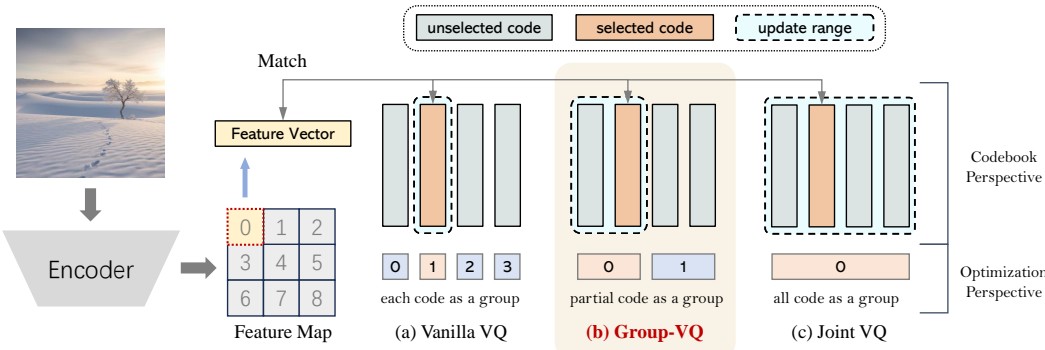

Figure 1: The differences among Vanilla VQ (a), Group-VQ (b), and Joint VQ (c) lie in their codebook update strategies: Vanilla VQ updates codes independently; Joint VQ optimizes the entire codebook jointly; and Group-VQ updates groups independently while optimizing jointly within each group.

multiple independent groups, where parameters are shared within each group to enable joint optimization within the group while keeping the groups independent of each other, i.e., group-wise optimization. This allows each group to focus on learning different feature distributions. Figure 1 shows a comparison among Vanilla VQ, Joint VQ, and Group-VQ. Additionally, we propose a codebook resampling method, which generates a new codebook by simply sampling after training. Our contributions can be summarized as follows:

- We propose the Group-VQ method, which approaches codebook optimization from a group perspective and balances the codebook utilization and reconstruction performance of the VQ model.

- We generalize the Joint VQ method based on shared parameters, and then introduce a post-training codebook sampling method to facilitate flexible adjustment of the codebook size without the need to retrain the model.

- We confirm the efficacy of Group-VQ and codebook resampling in image reconstruction, highlighting the importance of grouped design and outlining group number selection principles through ablation studies.

## 2 PRELIMINARY

The core of the visual tokenizer is vector quantization, which replaces any given vector with a discrete token from a codebook. For an image $I \in \mathbb{R}^{H \times W \times 3}$, VQ-VAE (Van Den Oord et al., 2017) first uses an encoder, typically a convolutional network with downsampling layers, to obtain the feature map $Z \in \mathbb{R}^{h \times w \times d}$, where $h$ and $w$ represent the height and width of the feature map, and $d$ is the number of channels in the feature map. The quantizer includes a codebook $C = \{q_i \mid q_i \in \mathbb{R}^d, i = 0, 1, \ldots, n-1\} \in \mathbb{R}^{n \times d}$, which means the codebook contains $n$ code vectors, each with a dimensionality of $d$. For each vector $z \in \mathbb{R}^d$ in the feature map, the quantizer finds the closest vector $q$ in the codebook (typically using Euclidean distance) to replace $z$.

$$q = \operatorname*{argmin}_{q_i \in C} \|z - q_i\|, \quad i = 0, 1, \ldots, n-1 \tag{1}$$

However, since the $argmin$ operation is non-differentiable, VQ-VAE uses the straight-through estimator (STE) (Bengio et al., 2013) to pass the gradient directly through $q$ to $z$, allowing the encoder to receive gradients from the reconstruction loss:

$$z_q = z + \mathrm{sg}[q - z] \tag{2}$$

Here, $sg[]$ denotes stop gradient, meaning that the gradient will not propagate through the $sg[]$ operation. By applying such quantization to each vector $z$ in the feature map $Z$, the feature map is converted into a discrete representation $Q \in \mathbb{R}^{h \times w \times d}$ composed of each $z_q$. The decoder then

generates the reconstructed image $\hat{I} \in \mathbb{R}^{H \times W \times 3}$ based on the quantized discrete feature map $Q$. The VQ-VAE loss consists of both the image reconstruction loss and the codebook loss:

$$\mathcal{L} = \|I - \hat{I}\|^2 + \beta\|Q - \text{sg}[Z]\|^2 + \gamma\|Z - \text{sg}[Q]\|^2 \tag{3}$$

where $\beta$ and $\gamma$ are fixed hyperparameters. However, a major issue with this approach is that in each iteration, only a subset of the code vectors are selected, meaning that only those vectors receive gradients and get updated, which eventually leads to codebook collapse (Roy et al., 2018; Huh et al., 2023; Yu et al., 2024b).

To achieve high codebook utilization, VQGAN-LC (Zhu et al., 2024a) proposes using a pre-trained visual backbone to extract image features and initializing the codebook $\hat{C}$ with cluster center vectors obtained through clustering. After initialization, the codebook remains frozen, and a projection layer $P(\cdot)$ is used to map the codebook to a projected codebook $C = P(\hat{C}) \in \mathbb{R}^{n \times d}$. During training, only this projection layer is optimized. This approach allows for the simultaneous adjustment of the entire codebook's distribution. LFQ (Yu et al., 2023) and FSQ (Mentzer et al., 2023) employ implicit and fixed codebooks to prevent codebook collapse. SimVQ (Zhu et al., 2024b) simplifies the aforementioned approach by directly employing random initialization for $\hat{C} \in \mathbb{R}^{n \times d}$ and reparameterizing the codebook $C \in \mathbb{R}^{n \times d}$ in the form of $\hat{C}W$, where $W \in \mathbb{R}^{d \times d}$. This method represents each code vector as a linear combination of the rows in $W$. By updating $W$, each code vector is indirectly updated. Huh et al. (2023) proposes adjusting each code vector $\hat{q}$ using shared global mean and standard deviation, defined as $q = c_{\text{mean}} + c_{\text{std}} \times \hat{q}$, where $c_{\text{mean}}$ and $c_{\text{std}}$ are affine parameters with the same dimensionality as the code vectors and are shared across the entire codebook. This approach simplifies the matrix $W$ to a diagonal matrix and adding a bias vector.

The commonality among the aforementioned optimization methods for VQ lies in their use of shared learnable parameters to reparameterize the entire codebook, thereby enabling joint optimization of the codebook space to mitigate the issue of codebook collapse. As such, we collectively refer to them as Joint VQ (achieved through shared parameters).

## 3 METHOD

### 3.1 OBSERVATION

Joint VQ is straightforward and effective in addressing the codebook collapse problem; however, it suffers from a significant issue: the gradients generated by all selected code vectors affect the entire codebook distribution. While this helps the codebook quickly adapt to the feature distribution generated by the encoder, it remains overly coarse-grained. We believe this may lead to potential interference between updates of different codes, making it difficult for the codebook to adapt to complex distributions. To validate this conjecture, we conducted small-scale image re-

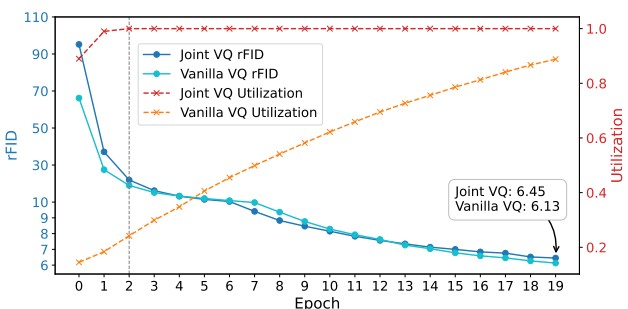

Figure 2: The codebook utilization rate and rFID for Joint VQ and Vanilla VQ at each epoch.

construction experiments comparing Joint VQ (implemented using SimVQ (Zhu et al., 2024b)) with Vanilla VQ, aiming to explore the relationship between codebook utilization and reconstruction quality. The parameter settings are detailed in Section 4.3.

The results are shown in Figure 2. It can be observed that Joint VQ rapidly achieves 100% codebook utilization by the 2nd epoch and maintains it, while Vanilla VQ's utilization grows gradually each epoch and ultimately remains below 90%. However, for the rFID metric (lower is better) used to measure image reconstruction quality, Vanilla VQ performs better in the end. We can thus infer that the performance gains of Joint VQ with larger codebook sizes stem from higher utilization, but the quality of the used codes is inferior to that of Vanilla VQ, which independently fine-tunes each code.

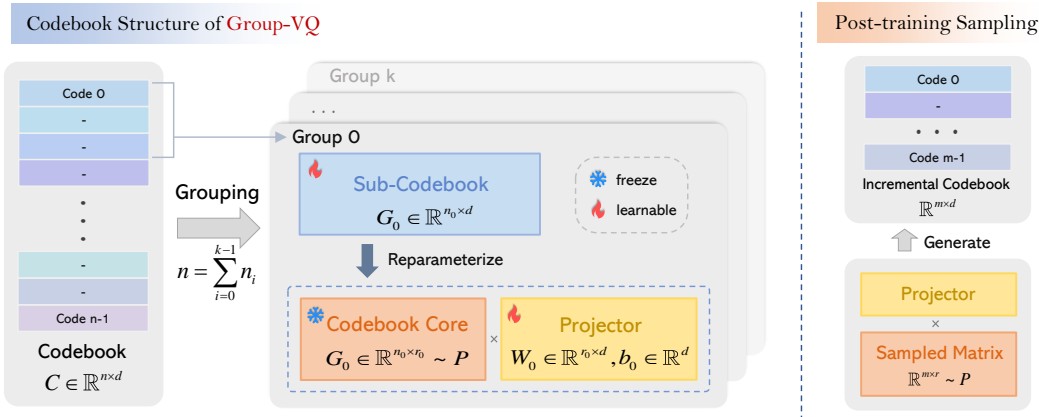

Figure 3: Left: The reparameterization method of the codebook by Group-VQ. It partitions the codebook into multiple disjoint groups (sub-codebooks), each of which undergoes separate reparameterization. Right: Post-training sampling. Simply sampling a new codebook core and applying the trained projector yields new codes.

## 3.2 GROUP-VQ

To address the trade-offs observed in our preliminary experiments, we propose a perspective on the codebook by considering it from the viewpoint of *groups*. In this context, a group is defined as the smallest unit of the codebook that is independently updated during training. Formally, let the codebook $C = \{q_i \mid q_i \in \mathbb{R}^d,\ i = 0, 1, \ldots, n-1\} \in \mathbb{R}^{n \times d}$ be partitioned into $k$ groups (or sub-codebook), where each group $G_j \subseteq C$ (for $j = 0, 1, \ldots, k-1$) contains $n_j$ code vectors. The groups are disjoint and collectively exhaustive:

$$\bigcup_{j=0}^{k-1} G_j = C,\ G_j \cap G_{j'} = \emptyset \text{ if } j \neq j' \tag{4}$$

Specifically, for a group $G_j = \{q_{j_1}, q_{j_2}, \ldots, q_{j_{|G_j|}}\}$, $q_{j_t}$ denotes the $t$-th code in $G_j$, and $\mathcal{L}_j$ represents the commitment loss corresponding to $G_j$. Since the groups are mutually disjoint and updated independently, when $j' \neq j$, $\mathcal{L}_{j'}$ does not depend on $q_{j_t}$, and thus $\nabla_{q_{j_t}} \mathcal{L}_{j'} = 0$. Therefore, the gradient of the total commitment loss $\mathcal{L}_{\text{cmt}}$ with respect to the code vector $q_{j_t} \in G_j$ is given by:

$$\nabla_{q_{j_t}} \mathcal{L}_{\text{cmt}} = \nabla_{q_{j_t}} \left( \sum_{j'=0}^{k-1} \mathcal{L}_{j'} \right) = \nabla_{q_{j_t}} \mathcal{L}_j + \sum_{j' \neq j} \nabla_{q_{j_t}} \mathcal{L}_{j'} = \nabla_{q_{j_t}} \mathcal{L}_j. \tag{5}$$

Consequently, during training, the update rule for each code $q_{j_t} \in G_j$ can be expressed as:

$$q_{j_t} \leftarrow q_{j_t} - \eta \nabla_{q_{j_t}} \mathcal{L}_j \tag{6}$$

where $\eta$ is the learning rate. The gradient $\nabla_{q_{j_t}} \mathcal{L}_j$ only affects the vectors within $G_j$, indicating that each group is updated independently based solely on the gradients computed from its contained codes, and the updates are confined to the vectors within $G_j$.

For each group $G_j$, in order to enable joint optimization of all code vectors within it, we can employ any method that supports joint optimization of codebooks. In Group-VQ, we adopt the parameter-sharing approach (Zhu et al., 2024b;a; Huh et al., 2023) to define each group. The group $G_j$ is parameterized as follows:

$$G_j = \hat{G}_j W_j + b_j \tag{7}$$

where $\hat{G}_j \in \mathbb{R}^{n_j \times r_j}$, $W_j \in \mathbb{R}^{r_j \times d}$ and $b_0 \in \mathbb{R}^d$. We refer to $\hat{G}_j$ as the "codebook core" and $W_j$ as the "projector". $r_j$ represents the rank of sub-codebook $G_j$. The values of $n_j$ and $r_j$ for each sub-codebook can be set differently, enabling the heterogeneity and asymmetry of the codebook. $b_j \in \mathbb{R}^d$ is the bias vector. Figure 3 (left) and Algorithm 1 illustrate the method to construct the codebook in the Group-VQ.

The vector quantization approach in Group-VQ remains unchanged. For each feature vector $z$ output by the encoder, the quantizer finds the closest vector $q$ in the codebook $C$. Since each feature vector always belongs to a specific sub-codebook, Group-VQ does not require an additional routing function design. Instead, feature vectors are automatically routed to the appropriate sub-codebook through distance metrics. Group-VQ inherently brings group-wise optimization during training.

**Discussion.** Under this group perspective, we can analyze existing vector quantization approaches. In Vanilla VQ, each code vector $q_i \in C$ is updated independently based on gradients from the feature vectors it quantizes, implying that each code vector forms its own group. This fine-grained update strategy results in a number of groups equal to the total number of code vectors, expressed as

$$k = n, \quad G_i = \{q_i\}, \quad \forall i = 0, 1, \ldots, n-1. \tag{8}$$

In contrast, Joint VQ, such as SimVQ (Zhu et al., 2024b) or VQGAN-LC (Zhu et al., 2024a), reparameterizes the entire codebook $C$ using shared parameters, meaning the update of any code vector affects the entire codebook, corresponding to a single group, described by $k = 1$, $G_0 = C$. The proposed Group-VQ balances these extremes by partitioning the codebook into $k$ disjoint and collectively exhaustive groups, where each group $G_j$ contains $n_j$ code vectors. This structure is formally defined as

$$C = \bigcup_{j=0}^{k-1} G_j, \quad G_j \cap G_{j'} = \emptyset \text{ if } j \neq j', \quad G_j = \{q_{j_1}, q_{j_2}, \ldots, q_{j_{n_j}}\}, \quad \sum_{j=0}^{k-1} n_j = n. \tag{9}$$

The core idea of our proposed Group-VQ is to balance the codebook utilization and its expressive power by adjusting the number of groups. If we consider the *code* as a whole and the *group* as the minimal unit, Group-VQ is equivalent to Vanilla VQ. Conversely, if we regard the *group* as a whole and the *code* as the minimal unit, each group becomes a Joint VQ. From a global perspective, Group-VQ represents a hybrid of these two approaches. By adjusting the number of groups $k$, where $1 \leq k \leq n$, Group-VQ achieves a hybrid of Vanilla VQ and Joint VQ, enabling flexible control over codebook utilization and expressive power. Appendix C illustrates the dynamics between the number of groups and codebook utilization rate.

### 3.3 CODEBOOK RESAMPLING AND SELF-EXTENSION

In VQ models, changing the size of the codebook typically requires retraining or fine-tuning the entire codebook along with the corresponding encoder and decoder. However, we note that this process is straightforward for generative codebooks, which refer to codebooks generated using a learnable network from a fixed distribution of codebook cores. After training, modifying the codebook size only requires resampling the codebook cores from the fixed distribution, while the model itself does not require further training.

Specifically, we define a generative network, also referred to as a projector $F_\theta(\cdot)$, and a codebook core $C_{\text{core}}$. Here, $F_\theta(\cdot)$ is a learnable linear projection layer or a small neural network, with $\theta$ denoting its learnable parameters. During the training process, each vector in the codebook core $C_{\text{core}}$ is randomly sampled from a fixed distribution $P$, with no learnable parameters. The final codebook $C$ is generated by applying $F_\theta(\cdot)$ to $C_{\text{core}}$:

$$C = F_\theta(C_{\text{core}}), \quad C_{\text{core}} \sim P \tag{10}$$

Since $C_{\text{core}}$ always follows the fixed-parameter distribution $P$, $F_\theta(\cdot)$ is trained to learn the transformation from the distribution $P$ to the feature distribution of the encoder's output. This property allows us to resample $C_{\text{core}}$ from the distribution $P$ after training, thereby flexibly adjusting the size of the codebook without incurring additional costs.

In Group-VQ, each sub-codebook belongs to the aforementioned generative codebook, so we leverage this property to apply it to post-training self-expansion of the codebook. Specifically, we randomly initialize each sub-codebook core $\hat{G}_j$, for example, by sampling each row vector $\hat{g} \in \mathbb{R}^{r_j}$ independently from a standard normal distribution $\mathcal{N}(0, I)$. After training, we sample new row vectors $\hat{v} \in \mathbb{R}^{r_j}$ from $\mathcal{N}(0, I)$ and project them using the already trained $W_j$ to obtain new code vectors:

$$\tilde{q} = \hat{v} W_j, \quad \hat{v} \sim \mathcal{N}(0, I) \tag{11}$$

These newly sampled code vectors $\tilde{q}$ are then added to $G_j$ to obtain the extended sub-codebook. The denser code vectors allow for finer quantization, thereby leading to improved reconstruction quality. Figure 3 (right) and Algorithm 2 illustrates the post-training sampling method.

**Algorithm 1** Codebook Initialization in Group-VQ

**Require:** Number of code vectors $n$, dimensionality $d$, number of sub-codebooks $k$ and sizes $\{n_j\}$, intrinsic dimensions $\{r_j\}$

**Ensure:** $\sum_{j=0}^{k-1} n_j = n$

1: **for** $j = 0$ **to** $k - 1$ **do**
2:     Initialize matrices:
3:       $\hat{G}_j \leftarrow \text{fix}(\hat{G}_j \in \mathbb{R}^{n_j \times r_j} \sim P)$
4:       $W_j, b_j \leftarrow$ random init
5:     $G_j \in \mathbb{R}^{n_j \times d} \leftarrow \hat{G}_j W_j + b_j$
6: **end for**
7:     $C \leftarrow \text{concat}(G_0, G_1, \ldots, G_{k-1})$
8: **return** the final codebook $C$

**Algorithm 2** Codebook Post-Training Sampling

**Require:** Trained $\{W_j, b_j, G_j\}$, target sizes $\{m_j\}$ for new sub-codebooks

1: **for** $j = 0$ **to** $k - 1$ **do**
2:     $G_j^\delta \leftarrow \hat{G}_j^\delta W_j + b_j$, $\hat{G}_j^\delta \in \mathbb{R}^{m_j \times r_j} \sim P$
3:     **if** Resampling **then**
4:       $G_j' \leftarrow G_j^\delta$
5:     **else if** Self-Extension **then**
6:       $G_j' \leftarrow \text{concat}(G_j, G_j^\delta[: m_j - n_j])$
7:     **end if**
8: **end for**
9:     $C' \leftarrow \text{concat}(G_0', G_1', \ldots, G_{k-1}')$
10: **return** the new sampling codebook $C'$

# 4 EXPERIMENTS

In Section 4.1, we demonstrate the superior image reconstruction performance of Group-VQ, including codebook resampling and self-extension experiments. Section 4.2 provides statistical analysis and visualization of sub-codebook information, along with experiments validating the group-wise optimization strategy. Section 4.3 investigates how to set the number of groups in Group-VQ for optimal performance through ablation studies.

## 4.1 VISION RECONSTRUCTION

**Implementation details.** For a fair comparison, we align the settings of SimVQ (Zhu et al., 2024b) and view it as a strong baseline. Specifically, we train Group-VQ with an input image resolution of $128 \times 128$. The image is processed by an encoder with multiple convolutional layers and downsampling layers, resulting in a total downsampling factor of $f = 8$. The encoder outputs a feature map of size $16 \times 16 \times 128$. For Group-VQ, the rank of each group is set to 128, and only each projector is trainable. We use the ImageNet-1k (Deng et al., 2009) and MS-COCO (Lin et al., 2014) datasets, with both VQGAN (Esser et al., 2021) and ViT-VQGAN (Yu et al., 2021) encoder/decoder architectures, and conduct different combi-

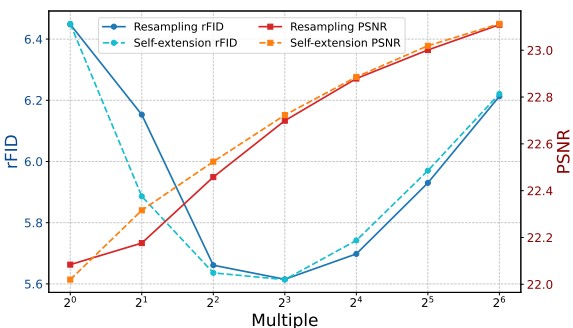

Figure 4: Using resampling and self-expansion methods, the rFID and PSNR under different multiples of expanding the codebook size. The codebook size during training is 1024.

nations to verify the broad applicability of Group-VQ. In our implementation, we simplify Group-VQ by evenly partitioning the codebook, ensuring an equal number of codes per group. So we use parallel implementation for faster speed and it can be found in Appendix B. All training is conducted on NVIDIA A5000 GPUs, with a batch size of 32 per GPU.

**Optimizer settings.** The learning rate is fixed at $1 \times 10^{-4}$, using the Adam optimizer (Kingma, 2014) with $\beta_1 = 0.5$ and $\beta_2 = 0.9$. We evaluate the performance of the Group-VQ method on the image reconstruction task using the rFID (reconstruction FID) (Heusel et al., 2017), LPIPS(VGG16) (Zhang et al., 2018), PSNR, and SSIM (Wang et al., 2004) metrics on the ImageNet validation set.

**Main results and analysis.** Table 1 presents the reconstruction performance of Group-VQ compared to other baseline methods. The primary baseline models include VQGAN (Esser et al., 2021), ViT-VQGAN, VQGAN-FC (Yu et al., 2021), FSQ (Mentzer et al., 2023), LFQ (Yu et al., 2023), VQGAN-LC (Zhu et al., 2024a), and SimVQ (Zhu et al., 2024b). The Group parameter in the table represents the number of groups in the codebook from the perspective of group-wise optimization. For

Table 1: **Reconstruction performance of various VQ models.** Group-VQ achieves the best reconstruction quality across all datasets and encoder/decoder architecture settings.

| Method | Codebook Size | Group | Codebook Usage | rFID↓ | LPIPS↓ | PSNR↑ | SSIM↑ |
|---|---|---|---|---|---|---|---|
| *Base Structure: VQGAN, Dataset: ImageNet-1k, Epoch: 50* | | | | | | | |
| VQGAN | 65,536 | 65,536 | 1.4% | 3.74 | 0.17 | 22.20 | 0.706 |
| VQGAN-EMA | 65,536 | 65,536 | 4.5% | 3.23 | 0.15 | 22.89 | 0.723 |
| VQGAN-FC | 65,536 | 65,536 | 100.0% | 2.63 | 0.13 | 23.79 | 0.775 |
| FSQ | 64,000 | 0 | 100.0% | 2.80 | 0.13 | 23.63 | 0.758 |
| LFQ | 65,536 | 0 | 100.0% | 2.88 | 0.13 | 23.60 | 0.772 |
| VQGAN-LC | 65,536 | 1 | 100.0% | 2.40 | 0.13 | 23.98 | 0.773 |
| SimVQ | 65,536 | 1 | 100.0% | 2.24 | 0.12 | 24.15 | 0.784 |
| SimVQ (ours run) | 65,536 | 1 | 100.0% | 1.99 | 0.12 | 24.34 | **0.788** |
| **Group-VQ** | 65,536 | 64 | 99.9% | **1.86** | **0.11** | **24.37** | 0.787 |
| *Base Structure: ViT-VQGAN, Dataset: 20% ImageNet-1k, Epoch: 40* | | | | | | | |
| ViT-VQGAN | 8192 | 8192 | 1.08% | 26.66 | 0.17 | 21.75 | 0.660 |
| LFQ | 8192 | 0 | 100.0% | 12.27 | 0.13 | 23.57 | 0.755 |
| SimVQ | 8192 | 1 | 100.0% | 11.44 | 0.12 | 23.74 | 0.761 |
| **Group-VQ** | 8192 | 8 | 100.0% | 10.72 | 0.12 | 23.85 | 0.764 |
| **Group-VQ** | 8192 | 16 | 100.0% | **10.67** | 0.12 | **23.87** | **0.765** |
| *Base Structure: VQ-GAN, Dataset: MS-COCO, Epoch: 20* | | | | | | | |
| LFQ | 4096 | 0 | 100.0% | 14.20 | 0.19 | 21.87 | 0.713 |
| SimVQ | 4096 | 1 | 100.0% | 13.18 | 0.18 | 21.90 | 0.719 |
| **Group-VQ** | 4096 | 16 | 100.0% | **12.55** | **0.17** | **22.08** | **0.722** |

Table 2: Codebook resampling and self-extension of Group-VQ on ImageNet-1k.

| Method | Codebook Size | Group | Codebook Usage | rFID↓ | LPIPS↓ | PSNR↑ | SSIM↑ |
|---|---|---|---|---|---|---|---|
| Group-VQ | 65,536 | 16 | 99.7% | 1.87 | 0.12 | 24.32 | 0.785 |
| + downsampling | 32,768 | 16 | 100.0% | 2.16 | 0.12 | 24.02 | 0.773 |
| + upsampling | 131,072 | 16 | 99.9% | 1.79 | 0.11 | 24.49 | 0.791 |
| + self-extension | 131,072 | 16 | 99.9% | 1.76 | 0.11 | 24.51 | 0.792 |

methods where each code is updated independently, such as VQGAN, the number of groups equals the codebook size. For methods with joint updates, the number of groups is 1. FSQ and LFQ, which utilize implicit and non-learnable codebooks, have a group number of 0. Group-VQ demonstrates state-of-the-art performance across multiple metrics, with improved rFID scores suggesting its reconstruction results align more closely with the original images' overall distribution. We present the images in Appendix E. In contrast, other methods with 100% codebook utilization exhibit varying performance. For instance, FSQ and LFQ completely freeze the codebook, resulting in relatively poorer performance. This suggests that we should not only focus on increasing codebook utilization but also enhance its learning capability.

**Codebook resampling and self-extension.** To validate the method proposed in Section 3.3, we resample the codebook to half the size used during training (downsampling) and double the size (upsampling) during evaluation. The self-extension method expands the codebook size by a factor of 2. The distinction between self-extension and resampling lies in the fact that the former retains the codes used during training, while the latter entirely replaces them with newly sampled codes. Table 2 reports the experimental results of codebook resampling and self-extension based on the post-training Group-VQ. The results demonstrate the expected decreases and increases in reconstruction metrics for downsampling, upsampling, and self-extension methods, respectively. Figure 4 presents additional results, where the codebook size during training is set to 1024, and it is expanded by powers of 2. PSNR consistently improves as the codebook size increases. We primarily focus on the more perceptually aligned rFID metric, which reaches its minimum value at an expansion of $1024 \times 2^3 = 8192$, before starting to rise. This indicates that codebook extension has a limit in scale: too many newly sampled and untrained codes can degrade reconstruction performance.

Table 3: Comparison of different codebook generation networks (projectors).

| Linear | MLP | Trainable Parameters | Codebook Size | Codebook Usage | rFID↓ | PSNR↑ |
|--------|-----|----------------------|---------------|----------------|-------|-------|
| ✓ | ✗ | 16,512 | 1024 | 100.0% | 6.45 | 22.02 |
| ✗ | ✓ | 33,024 | 1024 | 98.9% | 7.66 | 21.73 |
| ✓ | ✓ | 49,536 | 1024 | 99.7% | 6.66 | 22.15 |

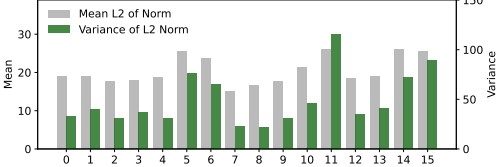
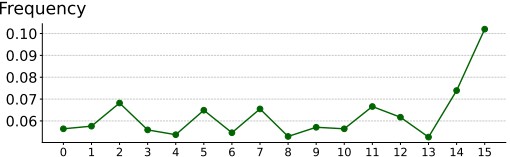

Figure 5: Left: The average value and variance of the $l_2$-norm of vectors in each group (0~15) in the codebook; Right: The usage frequency of each group when tested on the ImageNet validation set.

## 4.2 ANALYSIS OF GROUP-VQ

**Different groups have learned different patterns.** Analysis in the section is based on the Group-VQ with a codebook size of 65536 with a group number of 16 in Table 2. Figure 5 (left) shows the mean and variance of the $l_2$-norm of the codes in each group of the Group-VQ after training, and Figure 5 (right) shows the utilization rate of each group during the evaluation phase. These differences in the statistical values of the groups indicate that different groups have differentiated. Figure 6 shows the heatmap of the pairwise cosine similarity of codes. The checkerboard - like image indicates that the codes within a group are relatively similar, while the differences between groups are relatively large.

In Figure 7, we visualize the code vectors in a two-dimensional space using random projections (Johnson et al., 1986; Bingham & Mannila, 2001). In Figure 7(a), group 7 (with the smallest variance), group 11 (with the largest variance), and group 15 (with the highest usage frequency) exhibit distinct distributions. Group 15 shows a more dispersed distribution, corroborating its higher utilization rate. Figure 7(b) provides an overview of all groups. Figure 7(c) displays the bias vectors, i.e., the centers, for each group. More visualizations in D. Overall, different groups in Group-VQ adaptively learn diverse feature spaces, each responsible for distinct distributions.

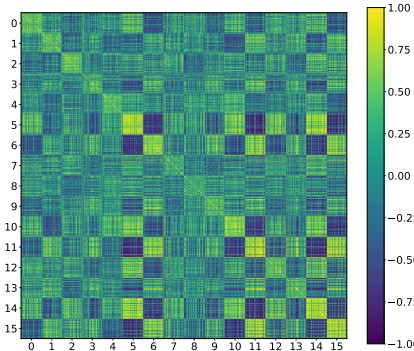

Figure 6: Heatmap of pairwise cosine similarity for codes. Randomly pick 128 codes from each group.

**Are more complex projectors effective?** The implementation of the Group-VQ method based on shared parameters is a reparameterization of the codebook. Therefore, it seems reasonable to consider that using a more complex network as the projector for generating the codebook might yield better results. We configure three types of projectors to verify whether this conjecture holds: the simplest linear projection layer, a multilayer perceptron (MLP) with one hidden layer, and a combination formed by summing the outputs of a linear projection layer and an MLP. We trained on a 25% subset of ImageNet for 20 epochs, with a codebook size of 1024 and the number of groups set to 1. Other hyperparameters and optimizer settings were the same as those in Section 4.1.

The experimental results, as shown in Table 3, indicate that the simplest linear projection layer achieved the best rFID, while the more complex MLP led to significant performance degradation. In the combination of a linear projection layer and an MLP, the two components seemed to counteract each other, resulting in performance that was intermediate between the two standalone approaches. This suggests that simply using a more complex projector network does not lead to better results and may even hinder the learning of the codebook. This result further underscores the necessity of independent updates for sub-codebooks in Group-VQ, indicating that its effectiveness stems from the use of multiple independent groups rather than a more complex codebook generation approach.

Table 4: The codebook utilization and image reconstruction performance under different settings of group numbers. A group size of 32~64 is considered optimal.

| | Group Count | | | | | | | | | | |
| --- | --- | --- | --- | --- | --- | --- | --- | --- | --- | --- | --- |
| | 1 | 2 | 4 | 8 | 16 | **32** | **64** | 128 | 256 | 512 | 1024 |
| Usage | 100% | 100% | 100% | 100% | 100% | 100% | 100% | 95.6% | 81.7% | 78.8% | 88.9% |
| rFID↓ | 6.45 | 6.52 | 6.29 | 6.19 | 6.13 | **6.05** | 6.09 | 6.11 | 6.15 | 6.28 | 6.13 |
| PSNR↑ | 22.02 | 22.07 | 22.12 | 22.13 | 22.13 | 22.14 | **22.16** | 22.08 | 22.02 | 22.06 | 22.14 |

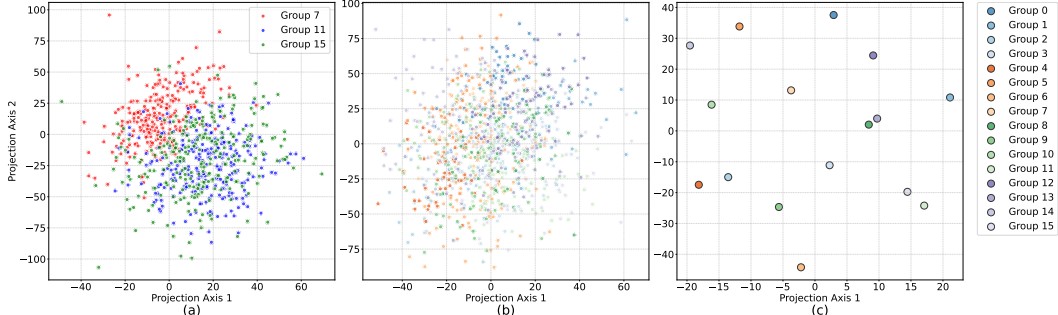

Figure 7: The visualization of the code vectors from Group-VQ with group=16 on ImageNet, randomly projected onto 2 dimensions. (a) shows 256 sampled points per group for groups 7, 11, and 15. (b) shows 64 sampled points per group for all groups. (c) shows the projection of the bias vectors for each group. This visualizes distinct distributions across different groups.

## 4.3 ABLATION STUDY

In this section, for efficiency, we conduct ablation experiments using a 25% subset of ImageNet. All models are trained for 20 epochs with a codebook size of 1024. Except for the number of groups in the codebook, all other parameters remain the same as that in Section 4.1.

Since each sub-codebook in Group-VQ is independently optimized, when the number of groups is set too high, it gradually degenerates into Vanilla VQ, leading to codebook collapse. Therefore, we need to carefully identify an appropriate number of groups to balance the modeling capability of the codebook while avoiding codebook collapse. Table 4 presents the codebook utilization and image reconstruction performance under different group settings. As can be seen from the table, when the number of groups increases from 1 to 32, the various metrics of Group-VQ generally show an improving trend. This indicates that, with a moderate increase in the number of sub-codebooks, a larger number of groups enhances the learning capability of the codebook without reaching the point of codebook collapse. However, when the number of groups increases further, the metrics of Group-VQ, particularly rFID, begin to rebound or fluctuate, and the codebook utilization starts to decline. The decrease in codebook utilization leads to wastage of the code, resulting in a gradual degradation of the metrics. Based on these experimental results, a group number of 32~64 is considered optimal. As the number of groups increases, more training time is required to achieve high utilization (shown in Figure 8). Given sufficient training time, a larger number of groups is the preferable choice.

## 5 CONCLUSION

In this paper, we first analyze the key differences between Vanilla VQ and Joint VQ. To balance codebook utilization and reconstruction performance, we propose Group-VQ, which introduces the idea of group-wise optimization of the codebook. We also propose a post-training codebook sampling method that enables flexible adjustment of the codebook size without retraining. In image reconstruction tasks, Group-VQ demonstrates superior performance. It would be interesting to explore the application of the group-wise optimization concept from Group-VQ to other, more complex Joint VQ methods in future work.

ETHICS STATEMENT

Our work introduces Group-VQ, a method for improving vector quantization in image reconstruction. While not directly enabling generative applications, VQ underpins such models, so we acknowledge potential misuse risks and advocate for responsible development. Public datasets may contain societal biases; we encourage future efforts to mitigate them. No human subjects or personal data were involved, and ethics guidelines are upheld.

REPRODUCIBILITY STATEMENT

To support the reproducibility of our research, we have provided an anonymous link to the core source code in the abstract: `https://anonymous.4open.science/r/Group-VQ_anonymous-60E3`. This code repository contains the core implementation of our proposed method along with configuration files. We encourage readers to consult this repository to obtain the complete technical details necessary to reproduce our results.

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

# A  RELATED WORK

Addressing the shortcomings of unstable codebook training and suboptimal encoding performance in Vanilla VQ models (Van Den Oord et al., 2017; Razavi et al., 2019), researchers have proposed various methods. The relevant work can mainly be categorized into: (1) Improvements to Straight-Through Estimator (2) Multi-index quantization (3) Improvements to the codebook.

**Improvements to Straight-Through Estimator.** Huh et al. (2023) alternately optimizes the codebook and the model, using the gradient of the task loss to update the codebook synchronously. Fifty et al. (2024) proposes a rotation trick to optimize the STE.

**Multi-index quantization.** The Vanilla VQ model represents each feature vector using a single code vector from the codebook, corresponding to one index. For an entire feature map $Z \in \mathbb{R}^{h \times w \times d}$, a total of $h \times w$ indices are used, which represents the theoretical upper limit of the information content after quantization. To increase the information capacity of the encoded representation, it is a natural idea to use more indices to represent the feature map, which also implies that more code vectors are selected and optimized. RQ-VAE (Lee et al., 2022) quantizes the error vector between the original and quantized feature vectors multiple times, allowing for a more precise representation of each feature vector. Product Quantization (PQ) (Jegou et al., 2010; Zhang et al., 2024; Li et al., 2024; Guo et al., 2024) divides the vector into multiple shorter sub-vectors and quantizes each sub-vector separately. VAR (Tian et al., 2024) introduces multi-scale residual quantization, where the feature map is downsampled multiple times, and each downsampled feature map is quantized. The commonality of these methods lies in the use of more than $h \times w$ indices to represent the feature map.

**Improvements to the codebook.** This section includes the way to look up code vectors and the optimization methods for the codebook. Van Den Oord et al. (2017) employs exponential moving average (EMA) to update the codebook. Sønderby et al. (2017); Roy et al. (2018); Kaiser et al. (2018) prevents certain codes from never being used by employing random sampling and probabilistic relaxation during the training process, while Dhariwal et al. (2020); Zeghidour et al. (2021) achieves this by periodically replacing inactive codes. The Vanilla VQ model uses Euclidean distance to measure vector distances, while VQGAN-FC (Yu et al., 2021) projects features into a low-dimensional space (Chen et al., 2024) and applies L2 normalization, making squared Euclidean distance equivalent to cosine similarity. DALL-E (Ramesh et al., 2021) utilizes Gumbel-Softmax trick (Jang et al., 2016) to represent the tokens of images. Some methods explore better codebook initialization, such as using features from a pre-trained backbone for K-Means clustering to initialize the codebook (Łańcucki et al., 2020; Huh et al., 2023; Zhu et al., 2024a). Lookup-free Quantization (LFQ) (Yu et al., 2023) and Finite Scalar Quantization (FSQ) (Mentzer et al., 2023) project feature vectors onto an extremely low dimension (typically < 10) and then perform integer quantization on each dimension respectively after compression by a bounded function. LFQ and FSQ are equivalent to fixing the codebook (Han et al., 2024; Zhao et al., 2024b), thus avoiding "codebook collapse". However, fixing the codebook leads to performance degradation, so there are various methods proposed to jointly optimize the entire codebook (Shi et al., 2024). Huh et al. (2023) perform affine transformations with shared parameters on each code. VQGAN-LC (Zhu et al., 2024a) and SimVQ (Zhu et al., 2024b) freeze the codebook and only train the projection layers after it. Zhang et al. (2023) aims to align the codebook distribution with that of encoder features.

# B   IMPLEMENTATION OF SIMPLIFIED GROUP-VQ

In the simplified Group-VQ, each sub-codebook has the same codebook size and the same rank. Consistent with the setup in Section 3.2, the codebook contains $n$ vectors, each with a dimension of $d$, and is divided into $k$ sub-codebooks. Each sub-codebook in the simplified Group-VQ contains $\frac{n}{k}$ (ensuring divisibility) code vectors, and all sub-codebooks have a rank of $r$. We use the following Algorithm 3 to parallelize the codebook generation.

---

**Algorithm 3** Codebook Initialization in simplified Group-VQ

---

**Input:** Codebook size $n$, code vector dimension $d$, number of sub-codebooks $k$, sub-codebook rank $r$

Random initialization (standard normal distribution): $\hat{C} \in \mathbb{R}^{\frac{n}{k} \times r}$, $W \in \mathbb{R}^{r \times (d \times k)}$, zero-initialized $b \in \mathbb{R}^{(d \times k)}$

Compute intermediate result: $C' = \hat{C} W + b$

Reshape $C'$ from $\left(\frac{n}{k}, d \times k\right)$ to $(n, d)$ to obtain the final codebook $C$

**Output:** Codebook $C$

---

Since the codebook core is fixed, the above method only parallelizes the codebook generation process of the simplified Group-VQ and does not affect the parameter independence between sub-codebooks. It is worth noting that in this simplified version, since each sub-codebook shares the same codebook core, performance may slightly degrade when the number of codes per group is too small (e.g., $<= 16$). However, the experimental setup in Section 4.1 does not fall into this range, so the impact can be ignored.

# C   CODEBOOK UTILIZATION DYNAMICS

Figure 8 illustrates that the codebook utilization rate increases with the progression of epochs. The greater the number of independently optimized groups, the later the 100% utilization rate is achieved.

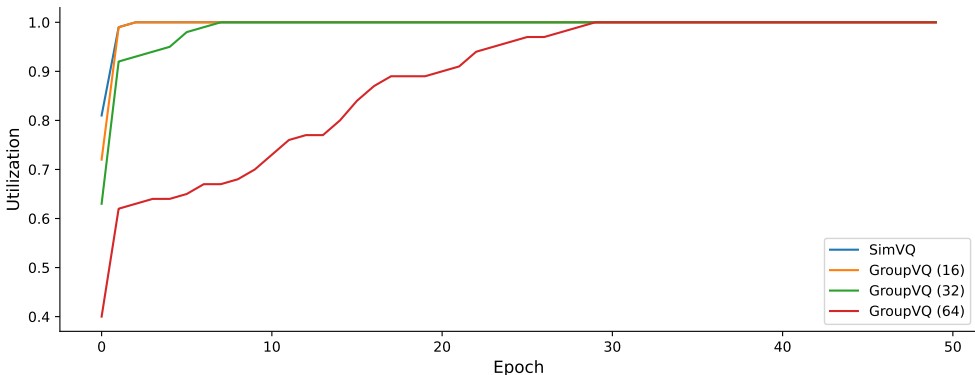

Figure 8: The codebook utilization rate increases with epochs. This includes SimVQ (equivalent to Group-VQ with a group number of 1) and Group-VQ with group numbers of 16, 32, and 64, respectively.

## D  MORE VISUALIZATION OF GROUP-VQ

In this section, we use Principal Component Analysis (PCA) (Pearson, 1901; Hotelling, 1933) to reduce the dimension of the codebook to two dimensions. We combine all the code vectors and standardize them. Then, we utilize PCA to extract the two principal components with the largest data variance, forming a projection matrix. All codes are mapped onto a two - dimensional plane through this projection matrix. The result is shown in Figure 9.

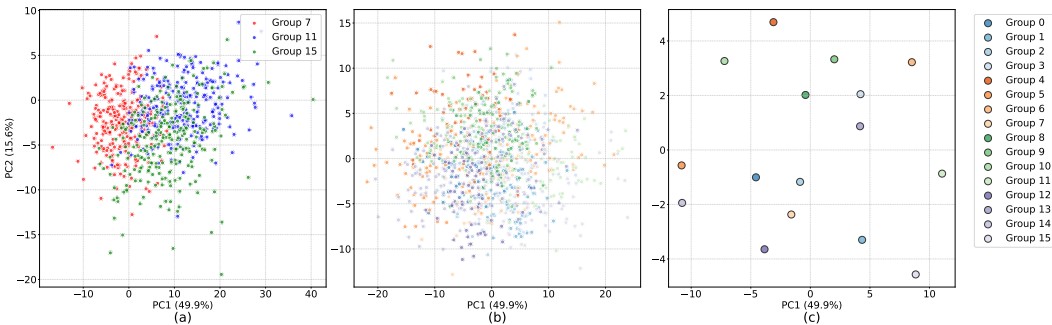

Figure 9: Use PCA to project the Group - VQ (group = 16) code vectors on ImageNet onto 2 - dimensional space for visualization. The horizontal and vertical axes are the first two principal components calculated by PCA. The horizontal axis represents the first principal component, PC1, which preserves the most information. The vertical axis represents the second principal component, PC2. The units of the coordinate axes are the values of the standardized data after being projected by PCA. (a) shows 256 sampled points per group for groups 7, 11, and 15. (b) shows 64 sampled points per group for all groups. (c) shows the projection of the bias vectors for each group.

# E SAMPLE RESULTS OF IMAGE RECONSTRUCTION

Figures 10 present a selection of sample results from the image reconstruction on ImageNet discussed in Section 4.1. For example, in the images of the fourth row, Group - VQ reconstructs the feathers at the tips of the bird's wings better. In the fifth row, the overall colors reconstructed by Group - VQ are more consistent with the original image.

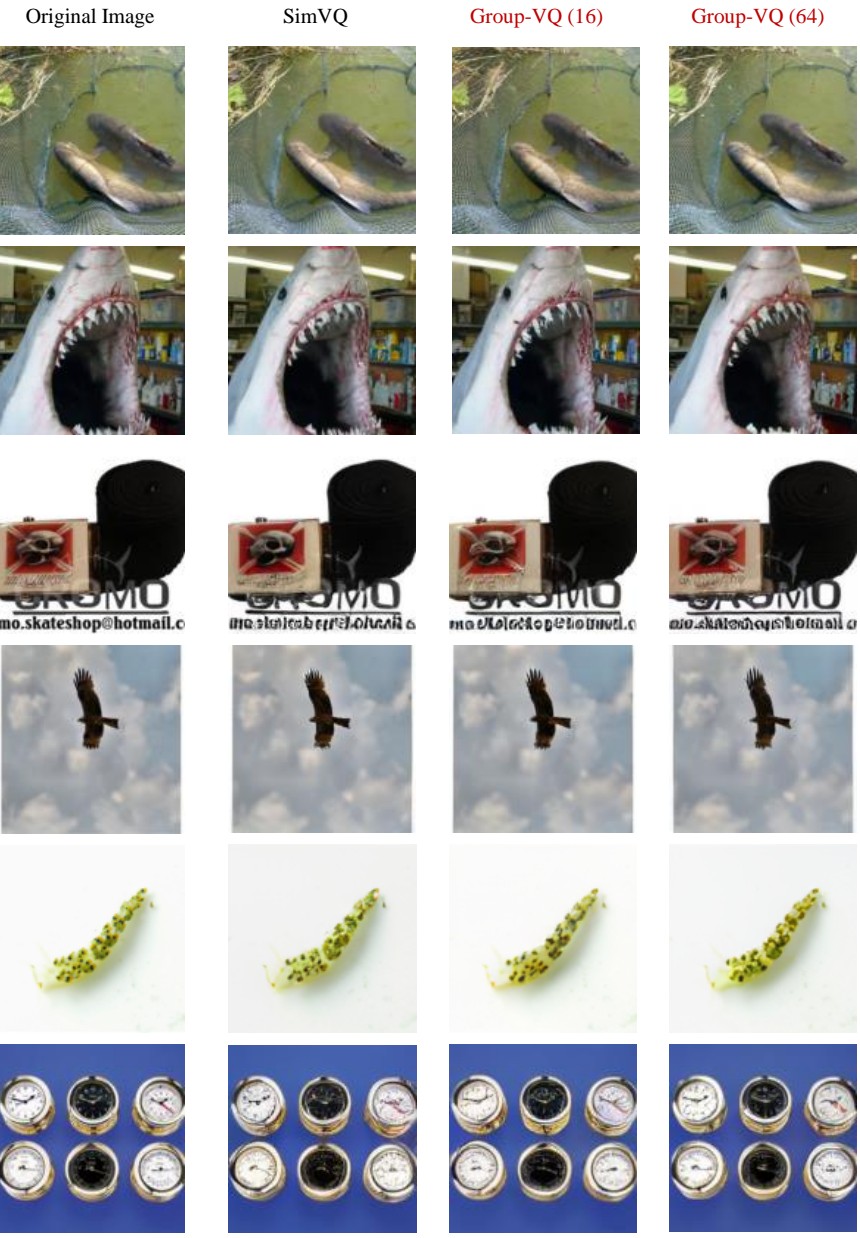

Figure 10: Image reconstruction examples. The numbers in parentheses for Group-VQ indicate the number of groups.

Figure 11 presents a selection of sample results from the codebook resampling and self-extension discussed in Section 4.1.

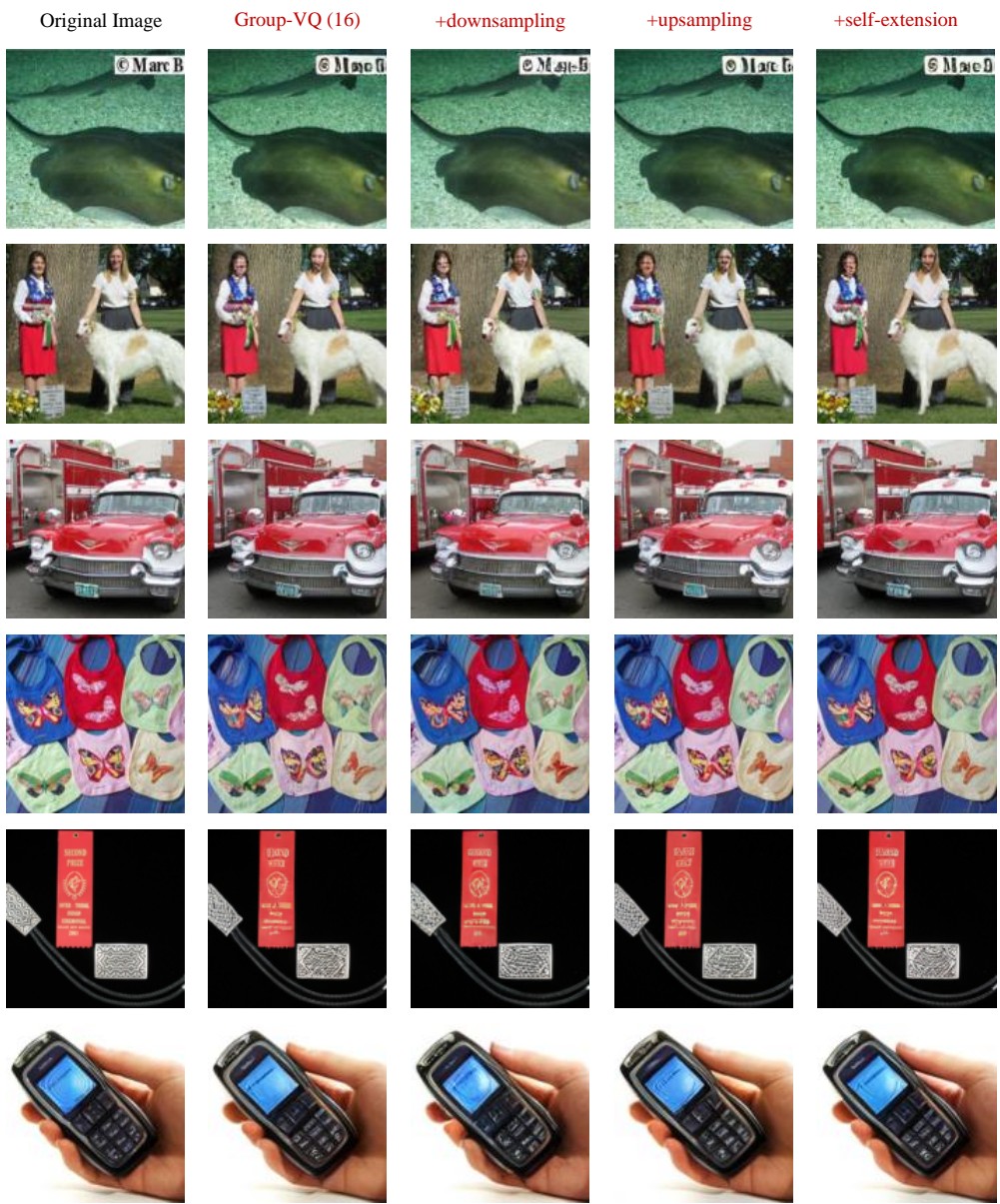

Figure 11: Images of codebook resampling and self-extension applied to Group-VQ with group=16.

## F  LIMITATION

The VQ model discretizes images into tokens, ultimately serving the modeling needs of downstream generative models. However, we have not yet validated the effectiveness of Group-VQ on generative models. We clarify that Group-VQ improves the optimization dynamics of the codebook during training solely by altering the parameterization of the codebook. Therefore, we believe Group-VQ remains orthogonal to downstream tasks, and thus we leave this exploration for future work.

## G  THE USE OF LARGE LANGUAGE MODELS

In the preparation of this manuscript, a Large Language Model (LLM) was used solely for the purpose of language polishing and stylistic refinement of the text. The LLM was prompted to improve clarity, grammar, and fluency of expression, without altering the core scientific content, methodology, results, or interpretations presented in the paper. The research ideas, experimental design, data analysis, and original writing were entirely conducted by the human authors. The LLM did not contribute to the generation of hypotheses, formulation of research questions, or development of novel concepts. Its role was strictly limited to post-writing linguistic enhancement.

