# OpenReview forum: "Group-Wise Optimization for Self-Extensible Codebooks in Vector Quantized Models"
_ICLR.cc/2026/Conference — ICLR 2026 Conference Withdrawn Submission_

### Official Review · Reviewer_oSUk · 2025-10-18

**Soundness:** 2
**Presentation:** 2
**Contribution:** 2
**Rating:** 2
**Confidence:** 4

**Summary:**

VQ-VAEs often suffer from codebook collapse, and "joint VQ-VAEs" have been proposed to improve the utilization of code vectors. The authors point out that increased codebook usage does not necessarily result in better reconstruction performance compared to vanilla VQ-VAEs. To address this, they propose a group-wise optimization of the codebook to balance codebook utilization and reconstruction quality. VQ-VAEs trained with this framework achieve better reconstruction performance than previous approaches and enable resampling and self-extension of the codebook.

**Strengths:**

- Bridging the gap between vanilla VQ-VAE and current joint VQ-VAEs offers an interesting perspective.
- The concept of enabling codebook upsampling is novel and intriguing.

**Weaknesses:**

- The most critical weakness is that the proposed method is not validated on any generation tasks. Application to generative tasks is important, as this is a primary use case for VQ-VAE in the vision domain (although VQ-VAE is also actively used as a neural codec in the audio domain).
- While Figure 2 is interesting problem posing, motivation of proposing Group-VQ is not so clear. It is not so straightfoward for me to think generalizing vanilla VQ and joint VQ with the concept of group is helpful.
- In Table 4, the improvement in reconstruction quality by varying the number of groups is marginal.
- Although self-extension and upsampling of the codebook are interesting, there is no comparison with results from Group-VQ trained from scratch with a doubled codebook size.
- The benefit of codebook resampling is unclear. Additionally, readers may wonder whether reconstruction performance improves if a fixed codebook core is used during training and inference.
- Figure 4 has no error bar despite that the resampling and self-extension involves random seeds.
- The section for related work lacks some works for "Improvement to the codebook" that is based on such as [1,2].

[1] Takida et al., ``SQ-VAE: Variational Bayes on Discrete Representation with Self-annealed Stochastic Quantization,'' ICML2022.

[2] Vuong et al., ``Vector Quantized Wasserstein Auto-Encoder,'' ICML2023.

**Minor**
- In Line 212, $b_0$ should be $b_j$?
- Visualizing similarities between projections for different groups, in addition to Figures 5 and 6, would be helpful.

**Questions:**

- Why do the trends of rFID and PSNR with respect to codebook size differ in Figure 4? I did not understand the interpretation of these results.

**Details Of Ethics Concerns:**

Potential ethic concerns are discussed in a section.

---

### Official Review · Reviewer_wmzt · 2025-10-21

**Soundness:** 2
**Presentation:** 2
**Contribution:** 1
**Rating:** 4
**Confidence:** 3

**Summary:**

They partition the VQ codebook into disjoint groups and reparameterize each group with its own projector/bias, so codes update jointly within a group but independently across groups (“Group-VQ”). Claimed benefit: balance high utilization with better reconstruction quality than “Joint VQ” (one shared projector for all codes) or “Vanilla VQ” (each code updated alone). They also propose a post-training “resampling/self-extension” trick to change codebook size without retraining.

**Strengths:**

1.Simple, modular design that interpolates between Vanilla and Joint VQ; the gradient-isolation argument for group-wise updates is clear
2.Practical knob (group count) with a reported sweet spot and utilization dynamics.

**Weaknesses:**

1. The paper frames Group-VQ as a middle ground between k=1 Joint VQ and k=n Vanilla VQ, but the core mechanism is “multiple independent shared-parameter blocks.” This is very close to simply sharding SimVQ/VQGAN-LC into independent sub-projectors—i.e., architecture factorization rather than a fundamentally new training principle. The paper needs a sharper theoretical or empirical claim that grouping yields qualitatively new optimization behavior beyond trivial partitioning.
2.Results show linear projectors outperform MLP projectors at same codebook size, but the study is brief and could reflect under-tuned MLPs or capacity/regularization mismatches. Without stronger controls (same params/FLOPs; weight decay; norm; init), it’s hard to attribute gains to “grouping” vs. “simpler projector works better here.”
3. While convenient, newly sampled codes aren’t trained; authors note quality eventually degrades when adding too many. This seems like a brittle knob unless coupled with light finetuning or usage-aware pruning. The paper should test a short finetune vs. pure resampling.

**Questions:**

1. If you fix total projector capacity (params/FLOPs), does grouping still win over a single global projector?
2. What happens if groups are data-driven (e.g., k-means over code usage) instead of uniform ranges? Any routing bias?
3. Does a brief post-resampling finetune (e.g., 1–5 epochs) recover the degradation when doubling/quadrupling codebook size?

---

### Official Review · Reviewer_ScTN · 2025-10-31

**Soundness:** 3
**Presentation:** 3
**Contribution:** 2
**Rating:** 4
**Confidence:** 4

**Summary:**

This paper proposes Group-VQ, a variant of VQ-VAE models designed to prevent codebook collapse and improve reconstruction performance.
The authors first show that JointVQ, a previous technique that addresses codebook collapse by using shared learnable parameters to reparameterize the entire codebook, has limitations in reconstruction performance.
They conjecture that sharing parameters across the entire codebook may lead to interference between code updates.
Based on this observation, Group-VQ divides the codebook into multiple independent groups, each optimized jointly within the group but independently across groups.
The paper also introduces a training-free codebook resampling technique, which allows flexible adjustment of codebook size after training.
Experiments on image reconstruction tasks demonstrate that Group-VQ outperforms existing methods, and that codebook resampling/self-extension is effective.

**Strengths:**

1. The paper is clearly written and the proposed method is well-motivated.
2. The experimental evaluation is comprehensive, including careful comparisons to strong baselines, extensive ablation studies on the number of groups and projector types, as well as detailed visualizations and analyses of the learned codebook.

**Weaknesses:**

1. The main idea, to my understanding, is dividing the codebook of SimVQ [Zhu et al., 2025] into groups and optimizing each group independently, and it feels relatively incremental. Grouping parameters in neural networks is a common strategy, as seen in grouped convolution [Krizhevsky et al., 2012], group normalization [Wu & He, 2018], and multi-head attention [Vaswani et al., 2017]. Thus, the novelty and impact of the work seem limited. This paper would benefit from a more explicit discussion of the unique advantages of Group-VQ over prior work, a clearer positioning with respect to existing literature, and the inclusion of theoretical guarantees.

2. While Section 4.1 shows that codebook expansion via self-extension/self-expansion is possible, it is unclear how effective the resulting codebook is compared to retraining. Results comparing against retraining Group-VQ and SimVQ or other codebook expansion methods would strengthen the claims.

Minor comment:
* The terms "self-expansion" and "self-extension" are used inconsistently throughout the paper. It would be better to unify the terminology for clarity (e.g., Figure 4 caption vs. Section 3.3).


[Zhu et al., 2025] Yongxin Zhu, et al. "Addressing representation collapse in vector quantized models with one linear layer." Proceedings of the IEEE/CVF International Conference on Computer Vision. 2025.

[Krizhevsky et al., 2012] Alex Krizhevsky, et al. "Imagenet classification with deep convolutional neural networks." Advances in neural information processing systems 25. 2012.

[Wu & He, 2018] Yuxin Wu and Kaiming He. "Group normalization." Proceedings of the European conference on computer vision (ECCV). 2018.

[Vaswani et al., 2017] Ashish Vaswani, et al. "Attention is all you need." Advances in neural information processing systems 30. 2017.

**Questions:**

1. Regarding Weakness 1: Could you provide further discussion about the unique advantages that Group-VQ offers over existing VQ approaches? In particular, how does Group-VQ fundamentally differ from these prior works, and are there specific scenarios where it provides clear benefits?
2. Regarding Weakness2: Do you have any comparisons between the proposed training-free codebook resampling techniques and other methods or retraining?

---

### Official Review · Reviewer_Pu5M · 2025-11-01

**Soundness:** 2
**Presentation:** 3
**Contribution:** 2
**Rating:** 2
**Confidence:** 4

**Summary:**

This paper introduces Group-VQ, an extension of Vector Quantized Variational Autoencoders (VQ-VAEs) that aims to mitigate the common problem of codebook collapse. Instead of using a single jointly optimized or static codebook, the authors propose a group-wise optimization strategy where the codebook is divided into multiple groups optimized independently but with intra-group coordination. This design seeks to enhance codebook utilization while maintaining high reconstruction quality. The paper also presents a training-free codebook resampling technique, enabling flexible post-training adjustment of codebook size. Experimental results on various image reconstruction tasks show that Group-VQ achieves improved reconstruction metrics and demonstrates adaptability through its resampling mechanism.

**Strengths:**

1. Novel Codebook Optimization Strategy:
The proposed group-wise optimization introduces a fresh perspective on improving codebook utilization in VQ-VAEs, effectively addressing the long-standing issue of codebook collapse.

2. Training-Free Flexibility:
The post-training codebook resampling technique is innovative and practical—it allows dynamic adjustment of codebook size without retraining, improving adaptability to different computational or performance requirements.

3. Potential Broad Applicability:
The framework could be easily extended to other domains (e.g., speech, video, or representation learning tasks) where quantized latent representations are useful.

**Weaknesses:**

1. The study is primarily inspired by the well-established VQ-VAE framework, and the authors introduce a new variant of the variational autoencoder. However, the absence of image generation results is a notable limitation of the paper.

2. The proposed approach shares conceptual similarities with rq-vae [1] and SC-VAE [2], as both employ multiple atoms or vector representations. Demonstrating improvements solely on image reconstruction tasks is insufficient; additional downstream evaluations are necessary to better highlight the advantages and generalizability of the proposed model.

[1] Lee, Doyup, et al. "Autoregressive image generation using residual quantization." Proceedings of the IEEE/CVF conference on computer vision and pattern recognition. 2022.

[2] Xiao, Pan, et al. "SC-VAE: Sparse coding-based variational autoencoder with learned ISTA." Pattern Recognition 161 (2025): 111187.

**Questions:**

Could you show image generation results? I am open to improve the score if the work demonstrates better image generation results compared to other vq-vae based models.

---

### Note · Authors · 2025-12-02

**Comment:**

We have carefully reviewed the reviewers’ feedback and sincerely appreciate the time and effort dedicated to evaluating our work.

While we firmly believe in the value of the proposed method, we also recognize the current limitations of the manuscript as highlighted in the reviews. We agree with the reviewers that further improvements are necessary. Therefore, we have decided to withdraw the current submission in order to thoroughly revise the paper and incorporate the constructive comments provided.

Thank you for your consideration.

**Withdrawal Confirmation:**

I have read and agree with the venue's withdrawal policy on behalf of myself and my co-authors.